# The Psychology of Murder Concealment Acts

**DOI:** 10.3390/ijerph18063113

**Published:** 2021-03-18

**Authors:** Mohammad Rahim Kamaluddin, Naji Arafat Mahat, Geshina Ayu Mat Saat, Azizah Othman, Ian Lloyd Anthony, Sowmya Kumar, Suzaily Wahab, Saravanan Meyappan, Balan Rathakrishnan, Fauziah Ibrahim

**Affiliations:** 1Centre for Research in Psychology and Human Well-Being, Faculty of Social Sciences and Humanities, Universiti Kebangsaan Malaysia, Bangi 43600, Malaysia; ifauziah@ukm.edu.my; 2Faculty of Science, Universiti Teknologi Malaysia, Johor Bahru 81310, Malaysia; 3Forensic Science Programme, School of Health Sciences, Universiti Sains Malaysia, Kubang Kerian 16150, Malaysia; geshina@usm.my; 4Department of Pediatrics, School of Medical Sciences, Universiti Sains Malaysia, Kubang Kerian 16150, Malaysia; azeezah@usm.my; 5Forensic Psychiatry Unit, Hospital Bahagia Ulu Kinta, Ipoh 30000, Malaysia; iandelloyd@hotmail.com; 6Criminology Department, Karunya University, Coimbatore 641114, India; sowmkmr@gmail.com; 7Psychiatry Department, Universiti Kebangsaan Malaysian Medical Centre, Cheras 56000, Malaysia; suzaily@ppukm.ukm.edu.my; 8Kuala Lumpur Magistrates Court (Criminal Division), Kuala Lumpur 50480, Malaysia; msaravanan@medac.gov.my; 9Faculty of Psychology and Education, Universiti Malaysia Sabah, Kota Kinabalu 88400, Malaysia; rbhalan@ums.edu.my

**Keywords:** murder, murder victim concealment act, psychological traits, postmortem burning, dumping

## Abstract

The escalating trend of murder victim concealment worldwide appears worrying, and literature does not reveal any specific study focusing on victim concealment amongst convicted male Malaysian murderers. Therefore, this study was aimed at investigating the psychological traits that may underlie the act of murder concealment in Malaysia via mixed method approaches. Male murderers (*n* = 71) from 11 prisons were selected via purposive sampling technique. In the quantitative analysis, a cross-sectional study design using the validated questionnaire was used. The questionnaire contained murder concealment variables and four Malay validated psychometric instruments measuring: personality traits, self-control, aggression, and cognitive distortion. The independent sample *t*-tests revealed the significantly higher level of anger in murderers who did not commit concealment acts (8.55 ± 2.85, *p* < 0.05) when compared with those who did so (6.40 ± 2.64). Meanwhile, the Kruskal–Wallis H test revealed that anger and the personality trait of aggressiveness-hostility significantly varied across the different groups of murder concealment acts (*p* < 0.05). The qualitative data obtained via the in-depth interviews revealed two important themes for the murderers to commit murder concealment acts: (1) fear of discovery and punishment and (2) blaming others. These findings discussed from the perspectives of the murderers within the context of criminology and psychology may provide the first ever insight into the murder concealment acts in Malaysia that can benefit the relevant authorities for crime prevention and investigation efforts.

## 1. Introduction

Murder and violent crimes are serious public health issues that erode well-being of a society and country. Being defined as ‘the act by which the death is caused is done with the intention of causing death’ (Section 300(a), Penal Code (Act 574) 2015), murder is the most vicious crime as well as the most serious of malum in se, claiming lives of almost half a million people with the global average of intentional homicide rate of 6.2 per 100,000 population in 2012 alone [1]. The murder act per se is an unlawful homicide with malice aforethought [2]. In the corpus of criminal jurisprudence, every single physical act (actus reus) of the perpetrator must be coupled with the intention or guilty mind (mens rea) to incriminate him with the said offence and the same concept applies for murder offence (actus reus non facit reum nisi mens sit rea) [3]. Death due to the voluntary act of the perpetrator and the intention of doing so are the two crucial criteria as required by the law [4]. To sustain a conviction for murder offence, it is germane for the prosecution to establish that a bodily injury is present; nature of such injury proved; exist intention to inflict that particular bodily injury—not accidental or unintentional; and lastly, such injury type is sufficient to cause death in the ordinary course of nature [5].

Among other reasons, the brutal action of murder is committed by murderers to evade the law, continue with their murderous schemes and as a warning to the potential victims. Due to its notoriety, murder is viewed as a malady, and an undesirable facet amongst civilized societies [6]. While the latest data remain unreported, the average of intentional homicide rate for Malaysia was recorded at 2.11 per 100,000 population for year 2013 [7]. Since a person who commits murder is punished by death in many countries including Malaysia (Section 302, Penal Code (Act 574) 2015), post-murder actions (e.g., murder concealment), which showcases a murderer’s level of brutality beyond the act itself, are generally the norm rather than the exception. It is significant to note that even the act of disposing of evidence of a crime of murder like destroying the dead body of the victim or other form of related evidence is a crime under the law, punishable with mandatory imprisonment up to 7 years and a fine (section 201, Penal Code (Act 574)).

It has been indicated that the precautionary acts are the ‘behaviors that offenders commit before, during or after an offence that are consciously intended to confuse, hamper, or defeat investigative or forensic efforts for the purposes of concealing their identity, their connection to the crime, or the crime itself” [8]. With the goal of avoiding convictions, crime scene staging that refers to the deliberate alterations of physical evidence at the crime scene (including murder concealment) to mislead the authorities or to wrongly redirect an investigation [8,9] has been usually discovered. While in the past, murder victims were commonly left at crime scenes in identifiable conditions, the pattern has changed within these few years whereby murderers often conceal their victims in ways that make identification a challenging task, hence, delaying justice [8]. For instance, back in the year 1993 at the state of Pahang, Malaysia, a mysteriously disappeared politician by the name of Dato Mazlan was discovered dead wherein his body was decapitated and cut into 18 pieces and buried in the premises of the three perpetrators [10]. Being the act or the attempt to obliterate the fact that a murder has been committed, murder concealment can range from non-concealment and non-disfiguring to complete concealment and total disfigurement. Although many researchers agree that precautionary acts by offenders are commonly discovered during crime investigations [8,9,11] psychological traits relating to the behaviors/actions of murder concealment remain unreported. Considering variations in population characteristics from one country to another, it is possible that the modus operandi may also vary, an assumption that requires clarification.

### 1.1. Common Methods of Murder Concealment

Canter and Almond [12] while reviewing the literature on arson and its strategies indicated that fire was used to cover up crimes. Such a general indication was further supported by Tumer et al. [13] via their retrospective analyses on 83 autopsy records in Ankara Group Administration of Council of Forensic Medicine (1 January 1998–31 December 2008), reporting 13 cases of probable postmortem burning of bodies to cover homicidal actions. Tumer et al. [13] interestingly also cited burying, burning and throwing corpses into water as means to cover up evidence of criminal actions. While reviewing autopsy cases at the Institute of Legal Medicine of Padova, Italy as well as pertinent literature, Matteis et al. [14] reported that multiple methods of concealment of corpses were used to cover-up homicides. Mohammad Rahim et al. [15] through their 5-year retrospective research of murder incidents in Peninsular Malaysia (based on reported cases by one of the local newspapers) indicated that postmortem burning and soil burial seem to emerge as the popular methods for murder concealment in Malaysia. Despite being secondary studies [12,13,14,15] and the fact that their conclusions can be presumptive, they provide basic ideas on possible murder concealment methods used by murderers (viz. arson and postmortem burning, as well as soil burial). Undoubtedly, obtaining the primary information (preferably via surveys and in-depth interviews of convicted murderers) may provide better insights with regards to the preference of murder concealment methods, as well as the underlying psychological traits attributable to such preference.

Arson, which is a subtype of fire setting [16] is defined as the intentional destruction of property for unlawful purposes [17] that may include murder concealments [12,18,19]. In the context of arson, the murder concealment method involves postmortem burning, in which by doing so the perpetrators believe that forensically pertinent evidence can be destroyed [9]. To execute postmortem burning, flammable and combustible materials (e.g., paint, and paper), as well as accelerants (e.g., kerosene) are frequently used due to their easy accessibility [13]. Accelerant, which refers to fuel (solid, liquid, or gas), is commonly used to initiate and/or increase the intensity or spread of fire in order to destroy physical evidence, hence, concealing a murder. Due to the fact that complete destruction of the human body can only take place during sustained burning at high temperatures (e.g., cremation), a single application of the easily evaporative accelerant on the body alone may not completely destroy all the evidence, nevertheless making the process of forensic examination more difficult. Therefore, murderers tend to use large amounts of accelerants in their attempts to eradicate all traces of evidence and to prevent identification of the murder victim [20]. The outrageous and sensational murder case of Altantuya, a Mongolian national in the Malaysian soil involved a very intricate and meticulous murder concealment methods wherein gun and explosives were used in the killing at a forested hilly area, resulting in the retrieval of only bones, hair strands and tissues by the police but the corpus delicti (corpse) was never found at all [21]. On an almost identical modus operandi, is the example of sustained postmortem burning which was tangibly observed in the murder case of the famous Malaysian cosmetics queen Dato’ Sosilawati Lawiya and her three aides, whereby there was unsuccessful DNA identification of the body remnants, attributed to the burning under extreme heat using zinc sheets and logs in a rubbish dump at an oil palm tree plantation farm. It is for the purposes of incineration of the bodies of the victims [22]. Moreover, because most arsonists that underwent psychiatric assessment had histories of substance use disorders (especially alcohol [16], the important role of such substances during the commission of murder concealment [20] can be expected.

While soil burial is a common legitimate method for disposing human remains for publicly recorded deaths [23], it is also a popular method used by murderers to conceal murders [15]. The main purpose of illegal burial is for immediate disposal of a body, and as such, remote and isolated areas like plantations and estates, landslides, cemeteries, a cave-in and farms are the commonly chosen sites, accountable for about 44% of recovered murder victims in Peninsular Malaysia [15]. However, because clandestine burials are usually shallow [24,25], the bodies may become vulnerable to differential decomposition, even with subtle variations in environmental conditions [25]. In addition to the fact that the discovery of illegal burial was often accidentally made by trespassers, farmers, hikers and hunters, Mohammad Rahim et al.’s [15] further reported that the bodies were recovered in burnt form, or chopped into several pieces, presumably to make identification of bodies difficult. In the context of crime scene investigation, the use of ground penetrating radar to locate clandestine burials of homicidal victims has been suggested [26,27,28]. Changes in soil compaction, subsidence, soil moisture, stratigraphic disturbances, decomposition gases, superposition and desiccation that are indirectly caused by the burial are important clues for indicating soil burial of bodies [29].

Murder coupled with mutilation of the corpse has been recorded since the ancient times, and this act is still persistent across the globe [30]. Mutilation-murder is defined as “those homicides where the offender tried to dismember the victim” [31]. Gunn and Taylor [32] emphasized that dismemberment is a relatively rare method whereby after killing the victim, the perpetrator uses a very sharp cutting weapon (e.g., saw or axe) to sever the limbs and cut the body into small pieces. The act of mutilation-murder can be classified into defensive, aggressive, offensive, and necromantic types [33] detailed below. While the defensive mutilation-murder relates to the motive of hiding, moving, getting rid of evidence or making identification of the victim difficult, aggressive mutilation-murder is associated with aggressively strong emotions. On the other hand, offensive mutilation-murder is known as the lust and nacro-sadistic murder, whereby dismemberment of the body epitomizes the cruelty of the act. As for necromantic mutilation-murder, the dismembered body parts are regarded as a trophy, symbol or fetish.

Gupta and Arora [34] while studying the profile of mutilation-murder in Himachal Pradesh, India reported that the defensive type was the commonest form of mutilation-murder, similar to that reported by an earlier study in Germany, Austria and Switzerland [35]. Gupta and Arora [34] reported that most of the bodies were recovered at secondary crime scenes (e.g., mud/rocky-bank of rivulets along national and state highways) with the primary aim of disposing/getting rid of them. They further indicated that several corpses were found chopped off using moderately heavy and sharp weapons and their dismembered torso parts were recovered from boxes lying in banksides of rivulets. The same authors also reported a murder caused by strangulation in which the body was later mutilated by incineration using accelerants and rubber of wheels. However, a study that was performed in Sweden reported the higher proportion and incidence of offensive mutilations than that of the defensive type [36]. Considering disparities among the populations, the fact those variations in psychological traits being the potential factor influencing the preference of mutilation-murder types and subsequently the murder concealment methods of the corpse cannot be ruled out.

### 1.2. Relevance of Disposal Pathways in Homicide Investigation

It has been indicated that the chosen methods and sites of murder concealment may potentially provide pertinent information about the nature of the crime, the criminal experience of the offender, his/her relationship with the victim [37], as well as the awareness about forensic science [38]. In the context of sexual murder, offenders with organized psychological characteristics tend to move the body after committing homicide. On the other hand, for older victims having conflicts with the offenders before the crime, concealment of their bodies by the offender is less likely [39]. Despite substantial efforts to profile the characteristics of offenders focusing on the manner by which the bodies were disposed or concealed [40], specific studies on the different contextual factors relating to the choice made by offenders to leave/conceal their victims after the murder and their preferred methods remain sparse. Therefore, having suitable offender profiling that scrutinizes crime scene characteristics to produce descriptive data pertaining to behaviors and personality of a potential perpetrator appears useful and pragmatic for narrowing the list of suspects, and subsequently solving the crime [41]. Although numerous explanations have been put forth to elaborate the nature of the murder concealment act, the psychological traits/reasoning that forms the basis of the act of concealment has been largely unexplored, particularly from the context of empirical evidence.

### 1.3. The Current Study

Taking into account the probable nexus of murder concealment and offender psychological characteristics that can vary among the different populations, and the fact that such data remain unreported for many Asian countries including Malaysia, this present study that evaluated such aspects acquires forensic consideration. Hence this current study was aimed at investigating traits and rationales that lead to murder victim concealment among Malaysian murderers. Moreover, using convicted male murderers from 11 prisons in Peninsular Malaysia, this study further examined the psychological traits that may influence the act of murder victim concealment, as well as the chosen methods. Utilizing the previously validated Malay psychometric instruments that evaluated four major domains (viz. personality traits, self-control, aggression and self-serving cognitive distortion (SSCD)), we developed the following specific hypotheses: (1) murderers that concealed their victims had significantly higher psychological mean scores for all the four domains than those who did not and (2) the distribution of psychological variables remained similar across the different types of murder concealment. Since psychological analysis among criminals is vital to understand the nature and causes for a particular offense, it is envisaged that the outcomes herein would offer meaningful contributions for profiling murderers who concealed their victims that may prove useful for crime scene investigations especially among an Asian sample. The findings also can offer significant inputs in terms of norms and cross-cultural differences among Asian samples in terms of murder concealment.

## 2. Materials and Methods

### 2.1. Study Design and Participants

Ethical clearance (USMKK/PPP/JEPeM [264.3(4)]) from the Human Ethical Committee of Universiti Sains Malaysia was obtained prior to the commencement of this present study. While none of the participants that participated in this present study were coerced in any way or rewarded for their involvement, the secrecy and anonymity of their identities were discretely maintained for ensuring validity of their responses. Hence, coding method instead of real name was used in data collection procedures. Written informed consent was obtained from each respondent prior to their participation. The mixed method research design comprised of both quantitative and qualitative approaches was used in this study involving 71 Malaysian male murderers from 11 prisons located within Peninsular Malaysia via purposive sampling technique during 2014 to 2016. The response rate was 71% as few declined to participate in this study. The inclusion criteria required that the participants be Malaysian males, aged 21 years and above, charged for murder (Section 302, Penal Code (Act 574) (2015)) ability to read Malay, as well as provided written informed consent. Considering the safety of the researchers and because highly aggressive offenders can be potentially harmful, the Prison Department of Malaysia only provided participants that were classified as low-risk inmates the study. On the other hand, inmates charged for culpable homicide not amounting to murder were excluded from this study.

Using a guided self-administered questionnaire, sociodemographic, as well as data on variables relating to murder concealment among participants were examined. Further assessments for measuring the four psychological domains (viz. personality traits, self-control level, aggression, and cognitive distortion) were performed. The qualitative phase was phenomenological in nature, whereby face-to-face in-depth interview sessions were carried out for collecting qualitative data from the identified participants who had concealed their respective victim(s).

### 2.2. Quantitative Approach

The self-administered questionnaire was comprised of two sections. The first section contained variables pertaining to murder victim concealment, while the second part consisted of four Malay validated psychometric instruments. The reliability values for all instruments used were also calculated for this study. The detail descriptions are provided below: 

Section one: This section examined the socio-demographic data of all the 71 participants. Following this, the participants were required to state the type of concealment method that they used (if any).

Section two: This section consisted of four Malay validated psychometric instruments detailed below:Zuckerman-Kuhlman Personality Questionnaire-M-40-Cross-Cultural (ZKPQ-M-40-CC): This instrument was the simplified version of the original ZKPQ-50-CC [42] that consisted of 50 items for measuring the Alternative Five Factor Model personality traits. Following the outcome of a validation study for the Malaysian populations reported by previous researchers [43], only 40 items were incorporated in the Malay version (ZKPQ-M-40-CC). The ZKPQ-M-40-CC assessed five types of personality traits viz. Activity (Act), Sociability (Sy), Aggressiveness-Hostility (Agg-Host), Impulsive Sensation Seeking (ImpSS), and Neuroticism-Anxiety (N-Anx). The internal consistency of each traits in (Cronbach’s Alpha) ZKPQ-M-40-CC for the criminal population were: Activity (0.76), Sociability (0.80), Aggressiveness-Hostility (0.79), Impulsive Sensation Seeking (0.78) and Neuroticism-Anxiety (0.84) [43]. In this present study, the reliability values for all the traits ranged between 0.75 to 0.80.Self-Control Scale (SCS-M): This instrument is the Malay version of the Self-Control Scale developed by Mohammad Rahim et al. [44]. The original SCS was provided by Grasmick et al. [45] to operationalize low self-control elements based on the General Theory of Crime introduced by Gottfredson and Hirschi [46]. In this study, SCS-M was administered as a unidimensional scale with 18 items. The scales were reverse coded so that high scores would indicate low self-control and vice versa. The overall internal consistency (Cronbach’s Alpha) for the criminal population was 0.80 [44]. In this present study, the reliability value of SCS-M was 0.82.Aggression Questionnaire (AQ-12-M): This instrument is the shorter version of the Aggression Questionnaire previously developed by Buss and Perry [47]. The AQ-12 comprised of 12 items [48] measuring four types of aggression (viz. Physical aggression, Verbal aggression, Anger, and Hostility) with three items for each subscale. In this present study, the Malay validated AQ-12 with an overall internal consistency (Cronbach’s Alpha) of 0.80 for the criminal population, as reported by Zaihairul Idrus, Nor Hafizah Nor Hamid and Geshina Ayu Mat Saat [49] was used. The reliability value of this scale in this study was 0.83.How I Think Questionnaire (HIT-M): This instrument is the Malay version of “How I Think” (HIT) Questionnaire initially developed by Barriga et al. [50]. The HIT-M instrument that measures four subscales of SSCD (viz. self-centered, blaming others, minimizing/mislabeling, and assuming the worst) was used in this present study. With six items for each subscale, the reported internal consistency (Cronbach’s Alpha) of HIT-M for the criminal population was 0.90 [51]. The overall reliability value of HIT-M in this study was 0.85.

### 2.3. Qualitative Approach

While 15 participants committed murder concealment acts, the interviews were terminated on the 9th participant due to the attainment of thematic saturation point i.e., when no new theme relating to the psychology of concealment act emerged. Considering that the primary goal of this qualitative phase was to explore in-depth understanding on the psychology of murder concealment act rather than making inferences or generalizations related to this act, having the sample size of 9 upon reaching the saturation point appears adequate. The interview protocol consisted of open-ended questions pertaining to murder victim concealment acts. Questions, such as “why did you conceal the victim?”, and “what are the factors that motivated you to conceal the victim?”; were explored in the interview. Additionally, several probing questions were raised based on the responses given by the participants. The questions in interview protocol were developed based on a focus group discussion involving five subject matter experts in the field of psychology and criminology. In addition, pertinent literatures were also referred during the constructions of questions for interview protocol.

### 2.4. Analyses Protocol

The information gathered was compiled into a set of systematic and computerized data, analyzed using the IBM Statistical Package for Social Sciences (SPSS) version 23.0. To summarize the socio-demographic information and murder victim concealment profiles, descriptive statistics were used. Using the Kolmogorov–Smirnov and Shapiro–Wilk tests, the normality of data was evaluated. The Independent sample *t*-test was performed to compare the means of variances in the psychological variables between the murderers who concealed their victims with those who did not. For comparing the medians of variances of the psychological scores across specific types of concealment methods Kruskal–Wallis H with pairwise comparison using Mann–Whitney U test was used. For inferring statistical significance, the level of significance (α) of 0.05 was chosen.

By employing the open coding technique, thematic analysis was utilized to analyze all the inputs obtained during interviews, a common method for qualitative research [52,53]. Thematic analysis was performed in this present study due to its ability to recognize, identify, and systematically retrieve the recurrent themes that emerged from the transcripts [54]. Moreover, thematic analysis can be useful to identify, analyze, and report patterns or themes from the data [53]. The entire qualitative analysis was carried out using the NVivo 10.0 software. The analysis of qualitative data began with manual transcription of the interview conversations into Microsoft word format. The transcribed verbatim were checked by two independent experts to maintain the accuracy level of transcription. The transcribed verbatim were systematically reviewed, grouped into themes and analyzed for content. For this, NVivo Software was used to facilitate the assessment of themes and subthemes.

## 3. Results

### 3.1. Quantitative Findings

The socio-demographic information obtained from the participants are presented in Table 1. Results revealed that majority of the participants were aged between 21–29 years old (62.0%) followed by those between 30–39 (19.7%), 40–49 (11.3%), 50–59 (4.2%) and 60–69 (2.8%), with the mean age of 29.94 ± 10.76 years. 29 (40.8%) of the 71 participants were Malays; Indians, Chinese and other ethnicities contributed 24 (33.8%), 17 (23.9%) and 1 (1.4%), respectively. It was further found that majority of the participants were Muslims (45.1%), unmarried (46.5%) and semi-skilled workers (59.2%) (Table 1).

#### Murder Victim Concealment and the Underlying Psychological Traits

Table 2 represents the murder concealment acts performed by the participants. While 15 participants (21.1%) responded that they committed murder concealment acts, the other 56 participants (78.9%) did not conceal, leaving the bodies intact at the crime scenes. Results also revealed that dumping (12.7%) and postmortem burning (8.4%) as the preferred murder concealment acts among Malaysian murderers (Table 2). Comparison of psychological mean scores between participants who committed murder concealment acts with those who did not is presented in Table 3. It was evident that participants who did not conceal their victims had significantly higher mean of anger scores (8.55 ± 2.85, *p* < 0.05) when compared with those that concealed their victims (6.40 ± 2.64). Considering this significant difference observed between the two groups, further comparisons of psychological scores among (a) participants that committed dumping and (b) postmortem burning as murder concealment acts, as well as (c) those who did not commit any murder concealment act were made using the Kruskal–Wallis H test, and the data are presented in Table 4. There were significant differences in the distribution of aggressiveness-hostility and anger among the three groups of participants (*p* < 0.05, Table 4), resulting in rejection of the null hypotheses; however, Kruskal–Wallis H test alone was insufficient to reveal the significantly differing group(s). Therefore, the subsequent pairwise comparison using the Mann–Whitney U test was performed for the two statistically significant variables (aggressiveness-hostility and anger), and the inference is provided in Table 5. The median score for aggressiveness-hostility was significantly the highest in participants that committed dumping for concealing their victims (25.00, *p* < 0.05), followed by participants who did not conceal (20.00) and those who committed postmortem burning (16.00). Significantly higher median score for aggressiveness-hostility also prevailed in participants that did not conceal with those of postmortem burning (*p* < 0.05). As for anger, significantly higher median score was observed in participants who did not conceal their victims (8.00, *p* < 0.05) when compared with those of postmortem burning (4.00); statistically significant difference was not observed between participants who did not conceal with those of dumping (*p* > 0.05) (Table 5).

### 3.2. Qualitative Findings

#### Admission for and Acts of Murder Victim Concealment

Results of the qualitative analyses obtained via in-depth phenomenology interviews that provided opportunities for participants to express their experience and views on murder victim concealment acts are presented in this section. Although 15 participants admitted that they committed murder concealment acts, thematic saturation point was achieved on the 9th participant whereby no new theme relating to the psychology of murder concealment act emerged. In this present research, two murder concealment acts (viz. dumping and postmortem burning) were identified. According to one of the participants, the murderous act was concealed by disposing the victim in an area that was perceived to be difficult to be accessed and searched. *“We brought the body and started seeking places to dispose the body. At last, we disposed it at a dark place. I think that was a swamp area”* (Participant 2). It was perceived that by disposing the body in a swamp, the crime as well as the identity of the victim would be less likely to be traced. Different ideas and the subsequent victim concealment acts were indicated by Participant 4. The participant mentioned: *“I gave ideas to my brother on how to conceal the body by throwing it into a manhole. I told my brother, if we throw the body into a manhole, it will be safer as the body will drift to other drains. Therefore, my brother started to cut the body into six pieces; head, two hands, and two legs. My brother and I entered the torso and two legs in a sack and disposed them into a manhole, meanwhile, the other two hands and the head, we burnt them”* (Participant 4). 

In addition, Participant 6 admitted to thinking out loud via self-dialogue, not knowing that his wife heard him talking about concealing the victim. He weighed the slower likelihood of being found out for murdering someone via cutting, disposing into a river, and burning. He indicated that *“I was thinking… how to conceal the body. I was asking myself whether to cut the body into pieces and throw them into the river or to dispose the body at a dumpster and burn using car tyres. But at the same time, I told myself, if I burn the body, sure there will be a lot of smoke, huge fire… and people can trace that burning spot. Also, I thought whether to bury the victim using cement and limestone. However, I could not do anything since my wife was listening to my self-conversation and I was shocked. I had to think of other plans”* (Participant 6).

Therefore, it appears that dumping was the common murder victim concealment act among Malaysian murderers. Rivers, lakes, manholes and large drains are common places where murder victims are recovered. In several cases, murder victims were attached with heavy objects like stones, a motorcycle and iron rods, presumably to prevent the corpse from floating during the decomposition stage. Pertinently, the thematic analysis yielded two themes (viz. fear of discovery and punishment, as well as blaming the victim) that prompted the participants to conceal their murder victims by justifying their acts as detailed below.

### 3.3. Justifications for Victim Concealment Act

#### 3.3.1. Theme One: Fear of Discovery and Punishment

It was evident that participants who concealed their murder victims had first thought about fear of discovery and legal punishment, motivating them to commit murder concealment acts. For example, Participant 2 was afraid and he mentioned that *“I was very afraid at that time. I knew that if I was charged, I would be charged under Section 302 of the Malaysia Penal Code which can lead to the death sentence by hanging. Then, I started to conceal the body by disposing it in a remote area. I concealed the body because I am afraid”.* In another example, Participant 4 claimed that his dominant emotion was fear and he mentioned that “*I was in a state of panic, fearful and afraid at that time. The only choice left was to conceal the victim’s body. Thus, I concealed it. What I can say is…I did (concealed) because I am scared*”. Similarly, Participant 5 indicated that *“I did this (concealment) because I did not know what else I should do. I was very scared and afraid that I would be caught for this. I know I can escape from this by concealing the body. So, without deliberating further, I started to conceal my victim”.* Participant 7 also indicated that *“I was too afraid at that time because I will be caught. Then, my wife suggested to conceal the body and I just followed her suggestion. I placed the body in a luggage bag and dumped it at a forested area. Hence, I thought concealing the body is the right choice at that time”.*

#### 3.3.2. Theme Two: Blaming Others

Blaming others emerged as the second theme for committing the victim concealment acts. Based on the interview sessions, one-third of the participants exhibited this form of cognitive distortion by blaming others for their concealment act. For example, Participant 1 mentioned that *“I had to conceal the victim’s body because I was threatened by other friends that if I don’t follow them, I might be a murder victim too on that day. For your information, we (my friends and I) planned the murder together, but I was the one who ended up concealing the body”*. Another participant stated that *“The suggestion to conceal the body was given by my friends. They said, I can escape if I conceal the body. I just followed their instructions and disposed the corpse by throwing it into a manhole and burning it. Because of them, I concealed the body”* (Participant 4). Meanwhile, Participant 7 blamed his wife for his action of concealment. He said that *“I listened to my wife at that time. She was the one who gave this idea to conceal. I would never have thought of it on my own. I always listened to my wife. So, when she gave the suggestion, I agreed to conceal the victim. She is the main reason for this act”.*

## 4. Discussion

The primary purpose of this present study was to investigate the psychological aspects that may lead to murder victim concealment as an act or attempt to obliterate the fact that a murder has been committed by destroying and/or preventing the identification of the murderers as well as victims [8,9]. Considering that this was the first study that reported about psychological traits in murder concealment acts among male Malaysian murderers, and because such aspects may differ among different populations, comparisons with previously reported data may appear inappropriate. Nonetheless, the nexus between quantitative and qualitative outcomes reported here may prove useful in adding rigor and breadth to the current body of knowledge, as well as explaining the complexity, and depth of the relevant psychological traits relating to murder victim concealment acts among convicted male Malaysian murderers. The data may enable criminologists to establish offender profiles for crime investigations in Malaysia, as well as in countries with similar demographic characteristics.

Results of this present study revealed that 21.1% of the participants committed murder concealment acts, indicating its popularity as means of body disposal among convicted male Malaysian murderers, and the acts of choice being dumping (12.7%) and postmortem burning (8.4%). Our findings are consistent with the findings made by Beauregard and Martineau [40] that not all murderers resorted to murder concealment acts and in light of forensic awareness, dumping and postmortem burning are the two common strategies to destroy physical evidence at crime scenes. Our findings corroborated well with the findings reported by Burton, McNiel, and Binder [16] that an act like fire-setting is partly attempted ‘to conceal criminal activity’ as well as ‘to express anger or vengeance’. The act of postmortem burning with the intention to cover up the homicidal action was also reported by Tumer et al. [13]. Likewise, Congram [24] and Gupta and Arora [34] reported the disposal of a murder victim in a shallow grave in the woods covered with lime and cement as well as six other victims at secluded places such as side banks of rivulets to taunt forensic investigations, respectively. Hence, exploring the psychological traits relating to the choice of murder concealment acts among murderers merits further investigation. 

Based on the quantitative outcomes, majority of the psychological domains/subscales evaluated did not significantly affect the murder concealment act or lack thereof, except for aggressiveness-hostility and anger. It was observed that participants who did not conceal their victims had significantly higher anger scores when compared with those that concealed their victims, consistent with the fact that aggressive behavior and violent offending are related with uncontrolled anger [55,56]. One possible explanation for this is linked to the nature of anger itself. The emotional component of aggression is conceived as impulsive, thoughtless, and driven by anger [47,57]. The feeling of anger may encourage participants to brutally harm the victims by killing and leaving the body at the crime scene. Leaving the victims at the crime scene is a manifestation of expressive murder that is often committed by those who are short-tempered and suffer from volatile emotions with high level of anger [58,59,60]. Such an indication is in agreement with that of Block and Block [58] who asserted that expressive murder is a form of murder that occurred due to expression, volatile emotions and psychological states. In general, individuals with emotional states such as anger, hostility and frustration are prone to commit expressive murders. In contrast, murderers who exhibited lower levels of anger are more likely to commit instrumental murders with proper plans to conceal their victims.

Moreover, results of this study are in agreement with that reported by Beareguard and Field [39], as reviewed by Chan [61]. They concluded that murderers with better awareness about forensic science as well as more organized in their psychological characteristics are more likely to conceal or move the bodies to another location, while those disorganized/sloppy murderers are more likely to leave their victims at the crime scene without concealment. Being more psychologically organized may involve the feelings of being less angry or having better ability to control anger when compared with those who did not conceal their victims. Hence, the proposition that the decision as well as act of concealing their murder victims were made when the murderers were less likely to be influenced by anger appears to be well supported by evidence. Interestingly, the fact that participants who committed body dumping had the highest score for aggressiveness-hostility when compared with those who did not conceal as well as postmortem burning, further studies for exploring this phenomenon among the murderers may prove necessary.

Considering the increasing number of murder cases in Malaysia, understanding the murder victim concealment acts from the perspectives of the perpetrators via in-depth interviews is crucial to provide insights pertaining to the intent (mens rea) and the planning element (actus reus) of the murderers. Therefore, in this qualitative phase, the participants were asked to describe their experiences regarding murder concealment acts. In this context, Marshall [62] asserted that “an appropriate sample size for a qualitative study is the one that adequately answers the research questions”. Because there were only 15 participants out of 71 murderers who committed murder concealment acts and since the saturation point had been successfully attained, the sample size of nine murderers interviewed here was considered as adequate.

The first theme that emerged from the thematic analysis was fear of discovery and punishment. According to Acorn [63], factors that are associated with the fact that criminals may not fear punishment before and while committing the crimes include (a) imagination, (b) immediacy, (c) agency, (d) desperation and (e) anger as described below. While imagination relates to the fact that the suffering of punishment experienced by other criminals does not make the criminal to imagine the severity of punishment, immanence refers to the lack of fear as the criminal believes that such punishment is not worth worrying because it is too remote. Moreover, the criminal believes that he/she is skilled enough to avoid punishment (agency), as well as he is in a state of despair that inhibits the fear of punishment (desperation). Lastly, the criminal may view himself or someone he cares about as having suffered injustice, thereby inhibiting his fear for punishment (angry). In this context Acorn [63] further elaborated that a criminal only developed the fear of punishment after the crime was committed e.g., when he/she started to think about what was done as concrete facts of the crime begin to emerge. This fact is consistent with the responses obtained in this present study that almost all the participants who concealed their victims claimed that fear appeared as the main factor that encouraged them towards their concealment act. It was noted that they were afraid of getting caught for their offense (murder) rather than feeling guilty for murdering someone, and the post-murder actions were not due to volatile emotions, such as anger. This fact was evident in this present study as those who killed due to expressive emotions exerted higher tendency to leave their victims at the crime scene without concealment. Hence, it can be construed that murder victim concealment acts are prone to take place in planned and instrumental murders compared to that of expressive murders triggered by sudden and intense anger. Although the fear of discovery of the crime and punishment under the law that shall ensue lingers greatly in the mind of the criminals, it is fallacious and absurd to think that the concealment of the murder act will exculpate them totally from the iron fist of the law due to the lack of evidence. Conviction in the court of law could be sustained purely on circumstantial evidence alone even without the discovery of the deceased person’s body. The classical judicial pronouncement in the case of Sunny Ang back in 1966 whereby the body of the victim, Jenny was never found but there was other overwhelming circumstantial evidence on record such as the dire need of the perpetrator for money relating to the motive issue, the travelling of the perpetrator with Jenny to an island for scuba diving and her disappearance thereafter. The circumstantial evidence is to be viewed on its cumulative effect of whether in totality, it leads to the irresistible conclusion on the guilt of the perpetrator [64]. Discovery of incriminating items such as the personal belongings of the victim, blood stain of the deceased on the perpetrator and DNA retrieved from it (sections 45, 46 and 51 Evidence Act 1950), sudden and otherwise inexplicable transition from a state of indigence and a consequent change of habits or a profuse or unwanted expenditure inconsistent with the position in the life of the perpetrator are some occasions that are extremely unfavourable to the supposition of innocence [65]. These are also relevant presumption of facts under the law and further corroborative evidence (section 114(a) Evidence Act 1950). Even the conduct of pointing to the places where incriminating items were discovered after police investigation and fleeing from the place of crime in a rush or consternation are relevant conduct to mirror motive, preparation, and previous or subsequent conduct of the perpetrator to the subject matter of the crime (section 8 Evidence Act 1950).

The second theme that emerged from the thematic analysis was “blaming others”. From the viewpoints of criminology and psychology, “blaming others” can be defined as a form of cognitive distortion that allows one to carry out their act [50]. Barriga et al. [50] emphasized that blaming others is a form of secondary cognitive distortion. In this present study, most participants who concealed their victims exhibited the “blaming others” trait. Based on their responses, the participants blamed their friends and tried to impose their errors upon others (Participants 2 and 4). Participant 2 blamed his friends for forcing him (due to their threats) to commit the murder concealment act, whereas Participant 7 blamed his wife for the act. These responses imply that murder victim concealment acts may also take place when more than one offender is involved although this is more prevalent in instrumental murders than that of expressive murders.

## 5. Conclusions

The quantitative outcomes generated herein reflect the empirical assessment of the psychological traits that influenced the concealment act among murderers. The qualitative outcomes, on the other hand, enhance one’s comprehension on why some murderers concealed their murder victims. Collectively, the results of this present study revealed that specific psychological markers, such as anger, fear and cognitive distortion (blaming others), underlie the concealment act among male Malaysian murderers. While the findings would add scholarly knowledge to the fields of criminology, victimology and psychology, the information gathered may provide the first ever insight into the context of murder concealment acts among Malaysians that can benefit the country in its efforts for crime prevention and investigations.

## 6. Limitations and Future Research

We wish to indicate that purposive sampling was used in the study because, due to safety issues, the Prison Department did not allow us to use any form of randomization. The participants were selected by the Prison Department after considering their stable state of mind and the willingness to cooperate. As such, generalization of the findings cannot be made. Having said that, the findings reported here may still be useful to provide the first ever insight into murder concealment acts performed by the Malaysian murderers. Moreover, replicating the same study model for other populations may prove useful for elucidating the cross-cultural determinants pertaining to murder concealment acts.

## Figures and Tables

**Table 1 ijerph-18-03113-t001:** Socio-demographic profiles of participants (*n* = 71).

Variables	*n* (%)
Age group (years old)	
21–29	44 (62.0)
30–39	14 (19.7)
40–49	8 (11.3)
50–59	3 (4.2)
60–69	2 (2.8)
Ethnic	
Malay	29 (40.8)
Chinese	17 (23.9)
Indian	24 (33.8)
Others	1 (1.4)
Religion	
Islam	32 (45.1)
Buddha	16 (22.5)
Hindu	19 (26.8)
Christian	4 (5.6)
Marital status	
Single	33 (46.5)
Married	24 (33.8)
Divorced/separated	11 (15.5)
Widower	3 (4.2)
Occupational status	
Not working	8 (11.3)
Semiskilled	42 (59.2)
Clerical-skilled	9 (12.7)
Self-employed/business	8 (11.3)
Government servant	4 (5.6)

**Table 2 ijerph-18-03113-t002:** Murder concealment acts performed by the participants.

Murder Concealment Act	*n* (%)
Dumping	9 (12.7)
Post-mortem burning	6 (8.4)
No murder concealment	56 (78.9)

**Table 3 ijerph-18-03113-t003:** Comparison of psychological mean scores between the between participants who committed murder concealment acts with those who did not (*n* = 71).

Domain	Subscale	Mean (SD)	Mean Difference (95% CI)	t-Statistics (df)	Statistical Significance
Personality traits	Activity	28.96 (5.67) ^1^ 30.80 (5.55) ^2^	−1.84 (−5.11, 1.44)	−1.12 (69)	*p* > 0.05
	Sociability	26.48 (4.39) ^1^ 26.73 (7.33) ^2^	−0.25 (−4.44, 3.94)	−0.17 (69)	*p* > 0.05
	Aggressiveness-Hostility	20.61 (7.63) ^1^ 21.53 (6.11) ^2^	−0.93 (−5.19, 3.34)	−0.43 (69)	*p* > 0.05
	Impulsive Sensation Seeking	21.86 (6.61) ^1^ 20.87 (6.10) ^2^	0.99 (−2.79, 4.77)	0.52 (69)	*p* > 0.05
	Neuroticism-Anxiety	18.29 (5.05) ^1^ 16.73 (7.04) ^2^	1.55 (−1.64, 4.75)	0.97 (69)	*p* > 0.05
Self-control	Self-control	49.57 (8.43) ^1^ 48.20 (5.88) ^2^	1.37 (−3.26, 6.00)	0.59 (69)	*p* > 0.05
Aggression	Overall aggression	29.75 (7.95) ^1^ 28.80 (10.94) ^2^	0.95 (−4.06, 5.96)	0.38 (69)	*p* > 0.05
	Physical aggression	7.77 (3.07) ^1^ 7.80 (3.76) ^2^	−0.03 (−1.90, 1.84)	−0.03 (69)	*p* > 0.05
	Verbal aggression	6.16 (2.40) ^1^ 6.53 (2.70) ^2^	−0.37 (−1.80, 1.05)	−0.52 (69)	*p* > 0.05
	Anger	8.55 (2.85) ^1^ 6.40 (2.64) ^2^	2.15 (0.52, 3.78)	2.64 (69)	*p* < 0.05
	Hostility	7.27 (2.84) ^1^ 8.07 (3.22) ^2^	−0.80 (−2.49, 0.89)	−0.94 (69)	*p* > 0.05
Self-serving cognitive distortion (SSCD)	Overall SSCD	53.86 (15.92) ^1^ 51.87 (15.50) ^2^	1.99 (−7.19, 11.17)	0.43 (69)	*p* > 0.05
	Self-centered	12.57 (5.39) ^1^ 12.60 (5.95) ^2^	−0.03 (−3.22, 3.17)	−0.02 (69)	*p* > 0.05
	Blaming others	15.00 (5.99) ^1^ 12.93 (4.48) ^2^	2.07 (−1.25, 5.38)	1.24 (69)	*p* > 0.05
	Minimizations	14.38 (5.17) ^1^ 13.33 (4.85) ^2^	1.04 (−1.92. 4.00)	0.70 (69)	*p* > 0.05
	Assuming worst	11.91 (4.68) ^1^ 13.00 (4.94) ^2^	−1.09 (−3.84, 1.66)	−0.79 (69)	*p* > 0.05

Independent Sample *t*-test was used to compare the psychological mean scores between participants who did not conceal their victims ^1^ with those who committed murder concealment acts ^2^. 15 participants committed murder concealment, while the remaining 56 participants left the bodies at crime scenes without concealment.

**Table 4 ijerph-18-03113-t004:** Comparison of medians of psychological scores among participants that committed dumping and post-mortem burning as murder concealment acts, as well as those who did not commit murder concealment.

Null Hypothesis (H_0_)	Statistical Significance ^1^	Decision
The distribution of Activity is the same across the categories of participants	*p* > 0.05	Do not reject H0
The distribution of Sociability is the same across the categories of participants	*p* > 0.05	Do not reject H_0_
The distribution of Aggressiveness-Hostility is the same across the categories of participants	*p* < 0.05	Reject H_0_
The distribution of Impulsive Sensation Seeking is the same across the categories of participants	*p* > 0.05	Do not reject H_0_
The distribution of Neuroticism-Anxiety is the same across the categories of participants	*p* > 0.05	Do not reject H_0_
The distribution of overall Aggression is the same across the categories of participants	*p* > 0.05	Do not reject H_0_
The distribution of physical aggression is the same across the categories of participants	*p* > 0.05	Do not reject H_0_
The distribution of verbal aggression is the same across the categories of participants	*p* > 0.05	Do not reject H_0_
The distribution of anger is the same across the categories of participants	*p* < 0.05	Reject H_0_
The distribution of hostility is the same across the categories of participants	*p* > 0.05	Do not reject H_0_
The distribution of low self-control is the same across the categories of participants	*p* > 0.05	Do not reject H_0_
The distribution of overall Self-serving cognitive distortion is the same across the categories of participants	*p* > 0.05	Do not reject H_0_
The distribution of self-centered is the same across the categories of participants	*p* > 0.05	Do not reject H_0_
The distribution of blaming others is the same across the categories of participants	*p* > 0.05	Do not reject H_0_
The distribution of minimizations is the same across the categories of participants	*p* > 0.05	Do not reject H_0_
The distribution of assuming the worst is the same across the categories of participants	*p* > 0.05	Do not reject H_0_

^1^ Kruskal–Wallis H test.

**Table 5 ijerph-18-03113-t005:** Comparison of median of aggressiveness-hostility and anger among participants that committed dumping and post-mortem burning as murder concealment acts, as well as those who did not commit murder concealment.

Psychological Variables	Groups			Statistical Comparison ^1^		
	No Concealment	Post-Mortem Burning	Dumping	No Concealment vs. Postmortem Burning	No Concealment vs. Dumping	Postmortem Burning vs. Dumping
Aggressiveness-hostility	20.00 (12.75)	16.00 (2.25)	25.00 (8.00)	*p* < 0.05	*p* < 0.05	*p* < 0.05
Anger	8.00 (5.00)	4.00 (5.50)	7.00 (4.00)	*p* < 0.05	*p* > 0.05	*p* < 0.05

^1^ Kruskal–Wallis H with pairwise comparison using Mann–Whitney U test was used. Data are presented as median (interquartile range).

## Data Availability

The data presented in this study are available on request from the corresponding author. The data are not publicly available due to data confidentiality imposed by Malaysia Prisons Department.

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
