# Peer review of "The Psychology of Murder Concealment Acts"

_ijerph, 2021, doi:10.3390/ijerph18063113_

Round 1

Reviewer 1 Report

Understanding the psychology of murder concealment is a very interesting topic as it can provide useful insights in the field of criminology, victimology and psychology.

In the introduction the authors explain which are the most common methods of murder concealment. They also explain that the chosen method of concealment can provide information about the nature of the crime, the offender's criminal experience and his or her relationship with the victim.

The aim of this research was to examine the psychological traits that may influence the concealment of murder as well as the methods chosen.

Perhaps the strength of this research lies in the mixed approach to the topic (quantitative and qualitative). This mixed perspective can serve to bring rigour and breadth to the body of knowledge and to explain in depth the features related to acts of concealment of murder victims among convicted Malaysian murderers.

The methodology used in the quantitative approach is well developed.

Among the main results, it is revealed that specific psychological markers such as anger, fear and cognitive distortion (blaming others) underlie the act of concealment among Malaysian murders.

Having analysed the manuscript, I would like to make a number of comments with the sole aim of improving the quality of the manuscript.

  1. The manuscript does not state when the study was conducted or when the sample was recruited. Please detail this when describing the sample.
  2. On the quantitative side, 4 psychometric instruments validated in Malay have been used, demonstrating acceptable internal consistency. However, you do not say how these tests performed in this particular population. Could you please detail the reliability of these tests in this particular sample?
  3. In the qualitative approach, a description is given of what the interviewed murderers think about the concealment of murder. It would be interesting if you could explain in a little more detail the methodology used, how the interview was structured, how the process of transcription of the interviews was carried out. On the other hand, it would also be interesting to carry out a content analysis in order to give more scientific rigour to the qualitative research.

  4. I detect that the bibliographical references are very old. Of the 65 references contained in the manuscript only 5 are less than 5 years old and 40 of them are more than 10 years old. Therefore, as far as possible, it would be necessary to use more up-to-date bibliographical references.
  5. Bibliographic references should be presented in Vancouver style as this is the most commonly used style in Health Sciences. Please adjust citations to Vancouver style and remove authors' names in the text whenever possible.
  6. Bibliographic citations should follow a logical order. For example [8, 9, 11]. It is embarrassing to see [9, 11, 8]. Please correct this detail in the text.
  7. When submitting an improved version of the manuscript, please make sure that the lines are numbered. It is very difficult to review with unnumbered lines.

Thank you.

Regards.

Author Response

Dear respected reviewer.

Attached herewith is the response to reviewer. Thank you sir. 

Reviewer 2 Report

The manuscript entitled “The Psychology of Murder Concealment Acts” presents interesting issue, but some areas must be corrected.

General:

The manuscript is shabbily prepared and it should be corrected (e.g. missing spaces, etc.)

The references should be prepared according to the instructions for authors.

Abstract:

Authors should present the aim of their study, not only what was done (e.g. “The aim of the study was…”)

“The findings are discussed from the perspectives of the murderers and within the context of criminology and psychology.” – instead rather the general conclusion should be formulated – e.g. something important for general knowledge, or society

Introduction:

This section is wordy and excessive, while it does not present a proper justification of the study.

Authors failed to justify the need for their study – they should present what is already known and what are the “gaps” in the scientific knowledge to formulate the aim of their study.

The aim should be briefly formulated

Materials and Methods:

The general characteristics of participants (Table 1) should be rather presented in materials section

Results:

Authors should verify the representativeness of the studied group

Authors should not reproduce in the text data that are already presented in tables

Tables should be stand-alone ones – be able to be understand without reading the manuscript, so Authors should explain everything needed in footnotes.

Tables which may be presented as a supplementary material should be transferred and not presented within the main body of the manuscript.

Discussion:

Authors should in their discussion include 3 areas: (1) compare gathered data with the results by other authors, (2) formulate implications of the results of their study and studies by other authors, (3) formulate the future areas which should be studied

Authors should present limitations of their study

Conclusions:

“As such, this present study adds useful knowledge to the fields of criminology, victimology and psychology” – instead rather the general conclusion should be formulated – e.g. something important for general knowledge, or society

Authors contributions:

It seems that contribution of some Authors was only minor (GAMS, AO) and they did not participate in preparing manuscript. There is a serious risk of a guest authorship procedure which is forbidden. In such case they should be rather presented in Acknowledgements Section and not be indicated as authors of the study.

Author Response

Dear respected reviewer.

Thank you for your valuable comments. Highly appreciated. 

Regards

Round 2

Reviewer 1 Report

Great improvements can be seen in this second version of the manuscript.

The authors have followed most of the recommendations provided by the reviewers and I believe that this version would be acceptable for publication.

I think that a good job has be done.

Congratulations!

Thank you very much

Regards